# TANGO1 recruits ERGIC membranes to the endoplasmic reticulum for procollagen export

António JM Santos[1,2†], Ishier Raote[1,2†], Margherita Scarpa[1,2], Nathalie Brouwers[1,2], Vivek Malhotra[1,2,3*]

[1]Centre for Genomic Regulation, The Barcelona Institute of Science and Technology, Barcelona, Spain; [2]Universitat Pompeu Fabra, Barcelona, Spain; [3]Institució Catalana de Recerca i Estudis Avançats, Barcelona, Spain

**Abstract** Previously we showed that membrane fusion is required for TANGO1-dependent export of procollagen VII from the endoplasmic reticulum (ER) (*Nogueira, et al., 2014*). Along with the t-SNARE Syntaxin 18, we now reveal the complete complement of SNAREs required in this process, t-SNAREs BNIP1 and USE1, and v-SNARE YKT6. TANGO1 recruits YKT6-containing ER Golgi Intermediate Compartment (ERGIC) membranes to procollagen VII-enriched patches on the ER. Moreover residues 1214-1396, that include the first coiled coil of TANGO1, specifically recruit ERGIC membranes even when targeted to mitochondria. TANGO1 is thus pivotal in concentrating procollagen VII in the lumen and recruiting ERGIC membranes on the cytoplasmic surface of the ER. Our data reveal that growth of a mega transport carrier for collagen export from the ER is not by acquisition of a larger patch of ER membrane, but instead by addition of ERGIC membranes to procollagen-enriched domains of the ER by a TANGO1-mediated process.

*For correspondence: vivek. malhotra@crg.eu

†These authors contributed equally to this work

## Introduction

Human cells express 28 collagens, most of which are secreted, constituting roughly 25% of our dry body weight (*Kadler et al., 2007*; *Malhotra et al., 2015*). Given their abundance and importance, an understanding of the mechanism by which collagens are sorted, packaged and exported across the secretory pathway, is of fundamental importance. A challenge arises from the fact that procollagens contain rigid, rod-like triple-helical domains that can reach 450 nm in length (*Burgeson et al., 1985*) and are too large to be exported from the ER by conventional COPII (Coat Protein complex II)-coated vesicles of 60–90 nm diameter (*Glick and Malhotra, 1998*; *Malhotra et al., 2015*; *Miller and Schekman, 2013*; *Saito and Katada, 2015*). Vesicles generated by a COPII-dependent mechanism have been characterised extensively (*Barlowe et al., 1994*; *Lee et al., 2004*) and it is clear that they are too small for procollagen export from the ER. So, how are collagens exported from the ER?

An important first step in addressing the mechanism of procollagen export from the ER was the identification of an ER-exit site resident protein called TANGO1 (*Bard et al., 2006*; *Saito et al., 2009*). The SH3-like domain of TANGO1 is required for its binding to procollagen VII in the ER lumen. On the cytoplasmic side, the second coiled coil domain of TANGO1 binds a protein called cTAGE5 and the C-terminal proline rich domains (PRD's) of TANGO1 and cTAGE5 interact with the COPII coat proteins Sec23A and Sec24C (*Saito et al., 2009*; *Saito et al., 2011*). Together, these proteins nucleate a complex that drives procollagen VII export from the ER (*Malhotra et al., 2015*; *Miller and Schekman, 2013*). TANGO1 mediates export of the only collagen expressed and secreted by Drosophila, and TANGO1 knockout mice are defective in the sorting and export of

many collagens (*Wilson et al., 2011*). Therefore, TANGO1 connects procollagens in the lumen of the ER to the cytoplasmic components of the COPII coats.

Other studies have suggested that ubiquitination of Sec31 could potentially increase the size of the COPII coats to produce a mega carrier for collagen export (*Jin et al., 2012*). Sedlin, a gene implicated in spondyloepiphyseal dysplasia tarda (*Christie et al., 2001*; *Matsui et al., 2001*; *Mumm et al., 2000*) that is characterised by altered extracellular matrix production, could modulate the size of nascent carriers by regulating Sar1A-mediated COPII coat dynamics for procollagen export from the ER (*Venditti et al., 2012*).

We showed previously that knockdown of Sly1 and the ER-localised t-SNARE, Syntaxin 18, inhibited the ER export of procollagen VII. This indicated that fusion of membranes with the ER was involved in events leading to procollagen VII export (*Nogueira et al., 2014*). We hypothesised that membranes from a post-ER membrane compartment fuse with a procollagen VII-enriched domain at the ER, which grows into a collagen VII-containing mega-carrier (*Malhotra et al., 2015*; *Nogueira et al., 2014*). We present here the entire complement of t-SNAREs required for procollagen VII export from the ER - BNIP1, USE1 and the previously described Syntaxin 18; that ERGIC membranes containing the v-SNARE YKT6 are required for fusion with the ER; and that the first coiled coil domain of TANGO1 is sufficient to recruit these membranes. We named this domain of TANGO1 TEER, for Tether of ERGIC at ER. The description of our findings follows.

## Results

### Membrane fusion is required for procollagen VII export from the ER

Membrane fusion requires four coiled coil domains collectively provided by v- and t-SNAREs (*Sutton et al., 1998*; *Weber et al., 1998*). We decided to identify all four SNAREs in the complex involved in procollagen VII export and, by localising the v-SNARE involved, identify the source of membranes in this fusion reaction. Syntaxin 18 (STX18), an ER-resident t-SNARE, and the SNARE-activity-promoting protein SLY1 are required for procollagen VII export, but not the t-SNARE Syntaxin 17 (*Nogueira et al., 2014*). Two other t-SNAREs reported to act in coordination with STX18 are BNIP1/Sec20 and USE1 (*Belgareh-Touze et al., 2003*; *Burri et al., 2003*; *Cosson et al., 1997*; *Dilcher et al., 2003*; *Lewis and Pelham, 1996*; *Sweet and Pelham, 1992*). siRNA oligos designed to knockdown t-SNAREs STX18, USE1 and BNIP1; or v-SNAREs YKT6, SEC22B and BET1 which function in transport between ER and the Golgi complex (*Hong, 2005*); and a scrambled oligo were transfected into RDEB/FB/C7 cells. 48 hr after transfection, cells were washed and re-plated in fresh medium containing ascorbate for 20 hr to promote procollagen export from the ER. The cell lysate and the medium from cells depleted of BNIP1, STX18, USE1, BET1, SEC22B or YKT6 and the control cells were probed for collagen VII. Depletion of t-SNAREs BNIP1, STX18 and USE1 resulted in approximately 91%, 72% and 78% block in collagen VII secretion compared to control cells, respectively. Depletion of v-SNAREs only revealed a significant reduction in collagen VII secretion for the case of YKT6 - approximately 84% (*Figure 1A,B*).

To identify the subcellular compartment in which collagen VII was arrested under these conditions, we immunostained RDEB/FB/C7 cells for collagen VII; and HSP47, calreticulin and TGN46 as markers of the ER and the Golgi complex, respectively. Depletion of the individual t-SNAREs led to intracellular accumulation of collagen VII that colocalised with HSP47 in ~74, ~59 and ~62% of the cells after depletion of BNIP1, STX18 and USE1, respectively, as opposed to ~8% measured in control cells. For the v-SNAREs, significant procollagen accumulation was only evident in cells depleted of YKT6 (*Figure 1C, D*). Taken together, these findings reveal that v-SNARE YKT6, and t-SNAREs BNIP1, Syntaxin 18, and USE1 are all required for the export of procollagen VII from the ER. Knockdown efficiencies were measured by RNA isolation followed by qRT-PCR to quantify the levels of remaining mRNA of USE1, BNIP1 and BET1; or by western blotting, to quantify remaining protein levels of STX18, YKT6 and SEC22B (*Figure 1E*). The effects of all these SNAREs on procollagen VII export were also confirmed qualitatively in Het-1A, a cell line that endogenously expresses collagen VII in a small fraction of the cells (*Figure 1—figure supplement 1*).

## YKT6 localises to the ERGIC

The v-SNARE YKT6 is a cytoplasmic protein that is recruited to membranes of the Golgi complex and the ERGIC (*Fukasawa et al., 2004*; *McNew et al., 1997*). We co-stained RDEB/FB/C7 cells for YKT6 and ERGIC-53 (a marker protein for ERGIC membranes) and observed several YKT6-positive vesicular structures, many of which colocalised with ERGIC-53 (*Figure 1F*) (Manders' overlap coefficient between YKT6 and ERGIC-53 is 0.705 ± 0.118). This confirmed the localisation of YKT6 to the ERGIC.

## TANGO1 recruits ERGIC-53-containing membranes to procollagen VII-enriched domains

We incubated RDEB/FB/C7 cells at 15°C to block exit of secretory cargoes from the ER, then shifted cells to 37°C for 5 min in the continued presence of ascorbic acid to initiate export of secretory cargo, including procollagen VII, from the ER. Cells were fixed and stained for ERGIC-53, collagen VII, and specific proteins that localise to the Golgi complex. We were struck by the location of ERGIC-53-containing membranes closely apposed to patches of procollagen VII in the ER (*Figure 2A* - first panel - and B). We then depleted the SNAREs involved in procollagen VII ER export and assessed the location of ERGIC-53 in relation to patches of procollagen VII in the ER. Depletion of the SNAREs revealed that ERGIC-53-containing membranes were closely apposed to these procollagen VII patches (*Figure 2A* - second to fifth panel – and B). Therefore, a non-SNARE constituent appears to tether ERGIC to ER domains.

To determine whether TANGO1 plays a role in recruitment of ERGIC-53-containing membranes to the ER, cells were transfected with a TANGO1-specific siRNA or a scrambled oligo. Western blotting of cell lysates for TANGO1 revealed 88 ± 5.8% reduction in the levels of TANGO1. As expected, depletion of TANGO1 arrested procollagen VII in the ER. Surprisingly however, ERGIC-53-containing membranes were displaced from the patches of ER that contained procollagen VII (*Figure 2A* - last panel – and B), suggesting that TANGO1 recruits ERGIC membranes to procollagen-enriched ER domains.

## A domain in TANGO1 directly recruits ERGIC membranes

As proteins involved in tethering are frequently characterised by coiled coil folds, we hypothesised that amino acids proximal to the ER membrane, including the first coiled coil domain of TANGO1 (TEER), could potentially tether and recruit ERGIC membranes to the ER (*Figure 2—figure supplement 1*). We first generated a Myc-epitope-tagged construct containing the full transmembrane domain of TANGO1 and the TEER domain (1177–1396 a.a. of TANGO1) (*Figure 3A*). As a control, we also generated a construct from TMCC3 (Transmembrane and coiled coil domains protein 3), a protein of unknown function, composed of its transmembrane domain and its first coiled coil domain (CC-TMCC3) (*Figure 3A*). After transfecting HeLa cells for 48 hr with the individual constructs, we immunostained cells for Myc and calreticulin to visualise the location of the expressed constructs and the ER, respectively. We found both TEER and CC-TMCC3 localised to the ER (*Figure 3B*). We then tested if the distribution of ERGIC-53 was affected by these constructs. HeLa cells expressing TEER, but not the CC-TMCC3, showed a significant colocalisation of ERGIC-53-containing membranes and the ER (*Figure 3C, D*), indicating that TEER is sufficient to recruit ERGIC membranes to the ER.

As shown above, the v-SNARE YKT6 is present in ERGIC-53-containing membranes, so after transfecting HeLa cells for 48 hr with the individual constructs we tested whether the membranes recruited by TANGO1's TEER domain contained YKT6. Since YKT6 is also present in the cytosol, we first permeabilised the cells and washed them extensively to remove cytosolic proteins, which was then followed by immunostaining for YKT6. This procedure revealed that YKT6-containing ERGIC-53 membranes were recruited by TEER to the ER, but not by CC-TMCC3 (*Figure 3E,F*).

## TEER recruits ERGIC membranes that contain YKT6, but not Golgi membranes or COPII coats

If the TEER domain was necessary and sufficient for recruitment of ERGIC-53 membranes to the ER, then its targeted expression at another compartment, for example the mitochondria, should lead to recruitment of ERGIC-53 membranes to mitochondria. Mitochondrial membrane-targeting domains

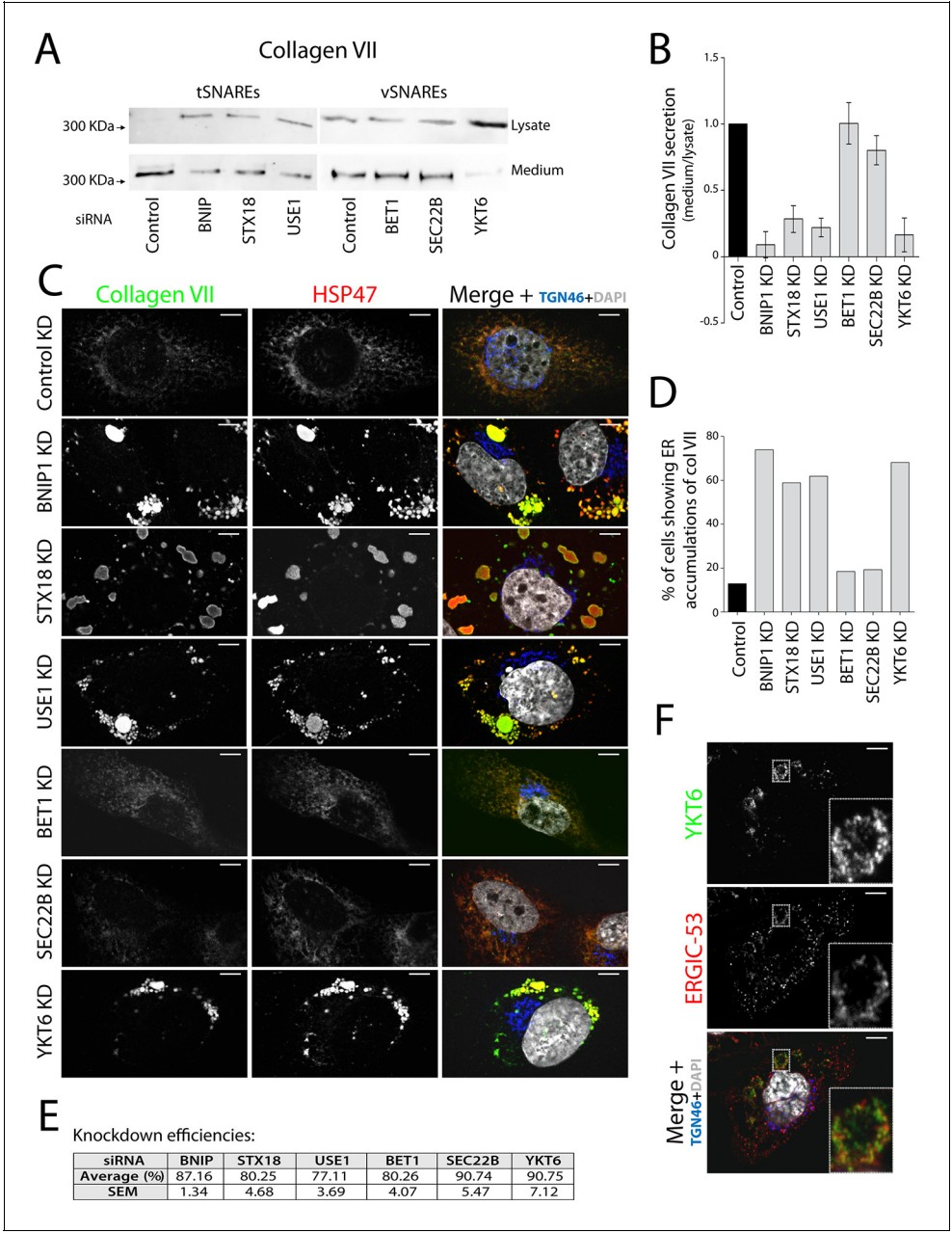

**Figure 1.** V- and t-SNAREs required for procollagen VII export from the ER RDEB/FB/C7 cells were transfected with siRNAs directed against BNIP1, STX18, USE1, BET1, SEC22B, YKT6 or a scrambled siRNA. (**A**) Collagen VII secretion was measured by western blotting and probing for collagen VII in RDEB/FB/C7 cell lysates and supernatants collected for 20 hr in the presence of ascorbic acid. In three independent experiments, intensities of the collagen VII signal in the lysate and the supernatant was recorded by densitometry. (**B**) The ratio of external vs. internal Collagen VII was normalized to quantify secretion in control cells as 1; Error bars: standard error of the mean (SEM). Differences in secretion between control and each knockdown were statistically significant, as determined by the Mann Whitney U test. (**C**) siRNA-treated RDEB/FB/C7 cells were seeded on coverslips and 20 hr after addition of ascorbate, cells were fixed and visualized with the indicated antibodies and DAPI by fluorescence microscopy (scale bars: 10 μm). (**D**) The percentage of cells that accumulate intracellular Collagen VII was determined by counting at least 20 cells in five random fields. (**E**) Efficiency of knockdown of each SNARE by quantifying mRNA or protein levels. (**F**) YKT6 is localised to vesicular structures, many of which colocalise with ERGIC-53. Manders' overlap coefficient 0.705 ± 0.118.

The following figure supplement is available for figure 1:

**Figure supplement 1.** SNAREs Syntaxin 18, BNIP1, USE1 and YKT6 are required for procollagen VII export from the ER in Het-1A cells.

are shorter than transmembrane domains of proteins inserted into the ER. We therefore generated a construct containing the TEER domain, but this time including only a short section of its transmembrane domain (residues 1188–1396 of TANGO1) tagged with a Myc epitope (*Figure 4A*). As a control, we generated a construct composed of the same short section of TANGO1's transmembrane domain (TANGO1's 1188–1197 a.a.) and the first coiled coil domain of TMCC3 (*Figure 4A*). 48 hr after transfection with the respective constructs, HeLa cells were fixed and visualised by immunofluorescence microscopy. Mitochondria were visualised using MitoTracker or an anti-ATP5A1 antibody. Our results revealed that both constructs localise to mitochondria (*Figure 4B,C*). To determine the topology of the mitochondrially targeted TEER (Mit-TEER) and CC-TMCC3 (Mit-CC-TMCC3), we performed a protease protection assay. A protein facing the cytoplasm is susceptible to proteolytic digestion, whereas a membrane encased protein is resistant in the absence of detergents. A crude mitochondrial membrane preparation was carried out from HeLa cells that were transiently transfected with either Mit-TEER or Mit-CC-TMCC3. Subsequently, these membrane preparations were incubated for one hour with proteinase K in the presence or absence of 1% NP40. Both Mit-TEER and Mit-CC-TMCC3 were degraded upon incubation of membranes with proteinase K in the absence of detergent; in contrast, ATP5A1, a subunit of the mitochondrial inner membrane-localised ATP synthase, was intact under the same conditions. As expected, inclusion of detergent further led to the degradation of ATP5A1 by proteinase K (*Figure 4D*). These data show that Mit-TEER and Mit-CC-TMCC3 are inserted into outer mitochondrial membranes and are exposed to the cytoplasm.

We then tested whether HeLa cells expressing Mit-TEER or Mit-CC-TMCC3 recruited ERGIC-53 membranes to the mitochondria. Our data revealed localisation of ERGIC-53-containing membranes to mitochondria expressing the TEER domain, but not to mitochondria expressing the CC-TMCC3 (*Figure 5A* - top and middle panels – and B). We could not detect any significant localisation of ERGIC-53 to mitochondria in untransfected cells (*Figure 5A* - bottom panel - and B). We also tested the ability of Mit-TEER to recruit YKT6-containing membranes. This was done by fluorescence microscopy after, as before, permeabilising and washing the cells to remove cytosolic proteins. HeLa cells expressing Mit-TEER, but not Mit-CC-TMCC3, recruited YKT6-containing membranes to mitochondrial surfaces (*Figure 5C* - top and middle panels - and D). We could not detect any significant localisation of YKT6 to mitochondria in untransfected cells (*Figure 5C* - bottom panel - and D). These results confirm the role of TANGO1's TEER domain in recruiting ERGIC-53- and YKT6-containing membranes.

Following a similar procedure, we tested whether Mit-TEER also recruited Golgi membranes (by staining the cells for Mannosidase II and TGN46) or membranes containing COPII coats (by staining for Sec31). Strikingly, neither Golgi membranes nor COPII coats were significantly associated with Mit-TEER, indicating that TEER specifically recruits YKT6-containing ERGIC membranes (*Figure 6A, C*).

It could be argued that Mit-TEER binds endogenous TANGO1 at ER-mitochondria contact sites (*Rowland and Voeltz, 2012*), leading to recruitment of YKT6-containing ERGIC membranes by endogenous TANGO1, but not by TEER per se. To address this possibility, we first transfected HeLa cells with Mit-TEER and observed the localisation of endogenous TANGO1. Immunofluorescence microscopy analysis revealed no significant association between endogenous TANGO1 and Mit-TEER (*Figure 6B* - left panel - and C), indicating that TEER at mitochondria does not interact with TANGO1 located at ER exit sites. In addition, we confirmed that there was no significant targeting of Mit-TEER to the ER (Manders' overlap coefficient between Myc and calreticulin is $0.306 \pm 0.05$). To further exclude any role of endogenous TANGO1 in this recruitment, we generated a HeLa cell line where endogenous TANGO1 was knocked out by CRISPR/Cas9 (ΔTANGO1) (*Cong et al., 2013*; *Mali et al., 2013*). Western blots confirmed complete depletion of TANGO1 in two different HeLa cell clones (*Figure 6—figure supplement 1A*) and as expected collagen VII secretion was severely inhibited (see methods, *Figure 6—figure supplement 1B*). Analysis by immunofluorescence microscopy revealed that Mit-TEER, when transfected in ΔTANGO1 cells recruited ERGIC-53-containing membranes (*Figure 6B* - right panel - and C). Together, these data strongly indicate that the TEER domain recruits ERGIC-53-containing membranes to regions of the ER decorated with TANGO1.

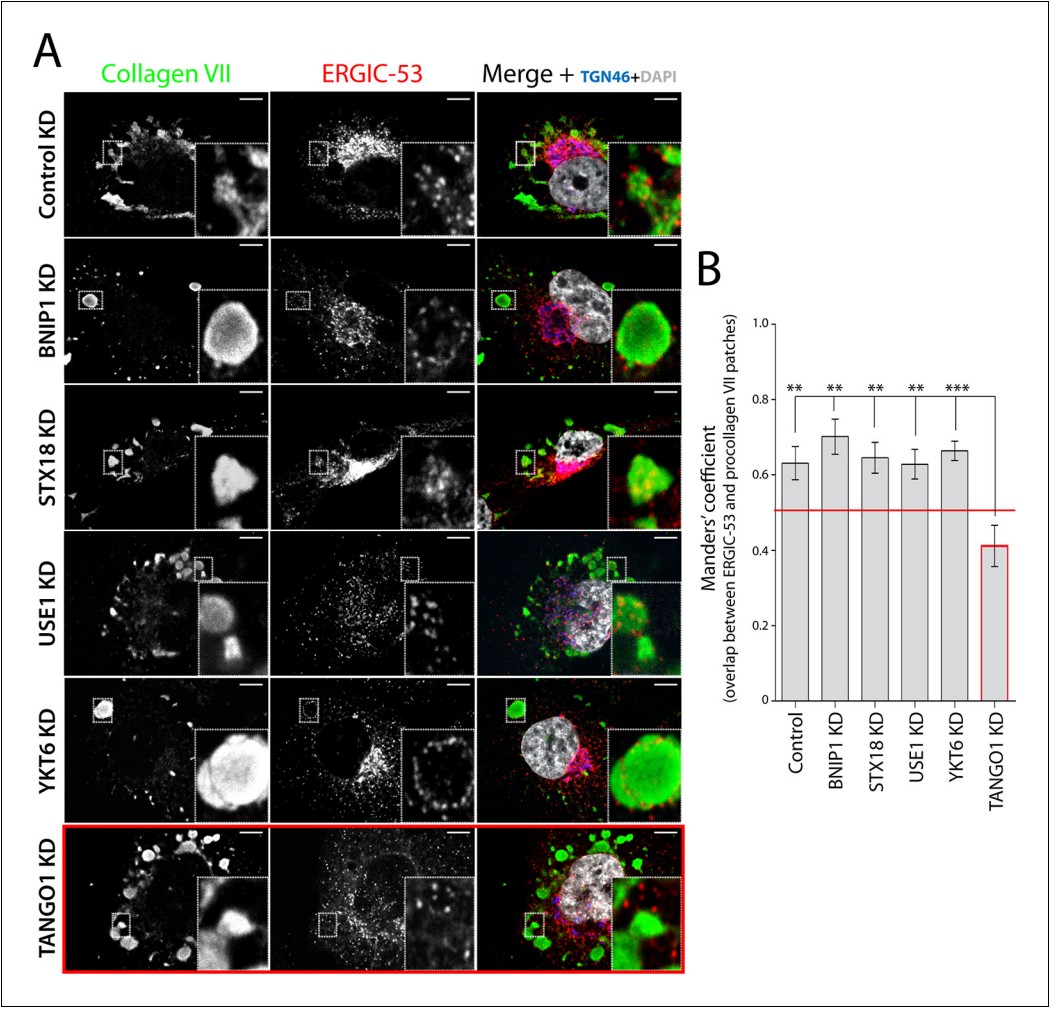

**Figure 2.** TANGO1-dependent recruitment of ERGIC membranes to ER-retained patches of procollagen VII. (**A**) Procollagen VII was accumulated in the ER of RDEB/FB/C7 cells that were either incubated at 15°C to accumulate secretory cargo in the ER and then released at 37°C for five minutes (control) or transfected with siRNA oligos to specifically deplete ER t-SNAREs BNIP1, Syntaxin 18 or USE1, the v-SNARE YKT6, or TANGO1. The cells were immunostained to monitor the localisation of ERGIC membranes and procollagen VII. Zooms are of 4x and scale bars correspond to 10 μm. (**B**) Colocalisation quantifications were calculated by Manders' correlation coefficient by measuring the overlap between the green and the red channels. Error bars reflect the standard error of the mean (SEM) of more than 20 cells from at least three independent experiments. **p<0.01, ***p<0.001.

The following figure supplement is available for figure 2:

**Figure supplement 1.** TANGO1 domains.

## Discussion

Our earlier studies indicated a seemingly counterintuitive requirement for the membrane fusion proteins SLY1 and STX18 in procollagen VII export from the ER (*Nogueira et al., 2014*). We have suggested that procollagen VII export from the ER requires fusion of membranes into the ER to generate mega-carriers. Unlike a standard COPII vesicle of 60–90 nm diameter that is moulded exclusively from the ER membrane by COPII coats, the growth of a mega-carrier for procollagen VII export depends on the addition of membranes from a non-ER source (*Malhotra and Erlmann, 2015*; *Malhotra et al., 2015*). The present study shows that the TANGO1 protein recruits ERGIC membranes that fuse to the ER for procollagen export.

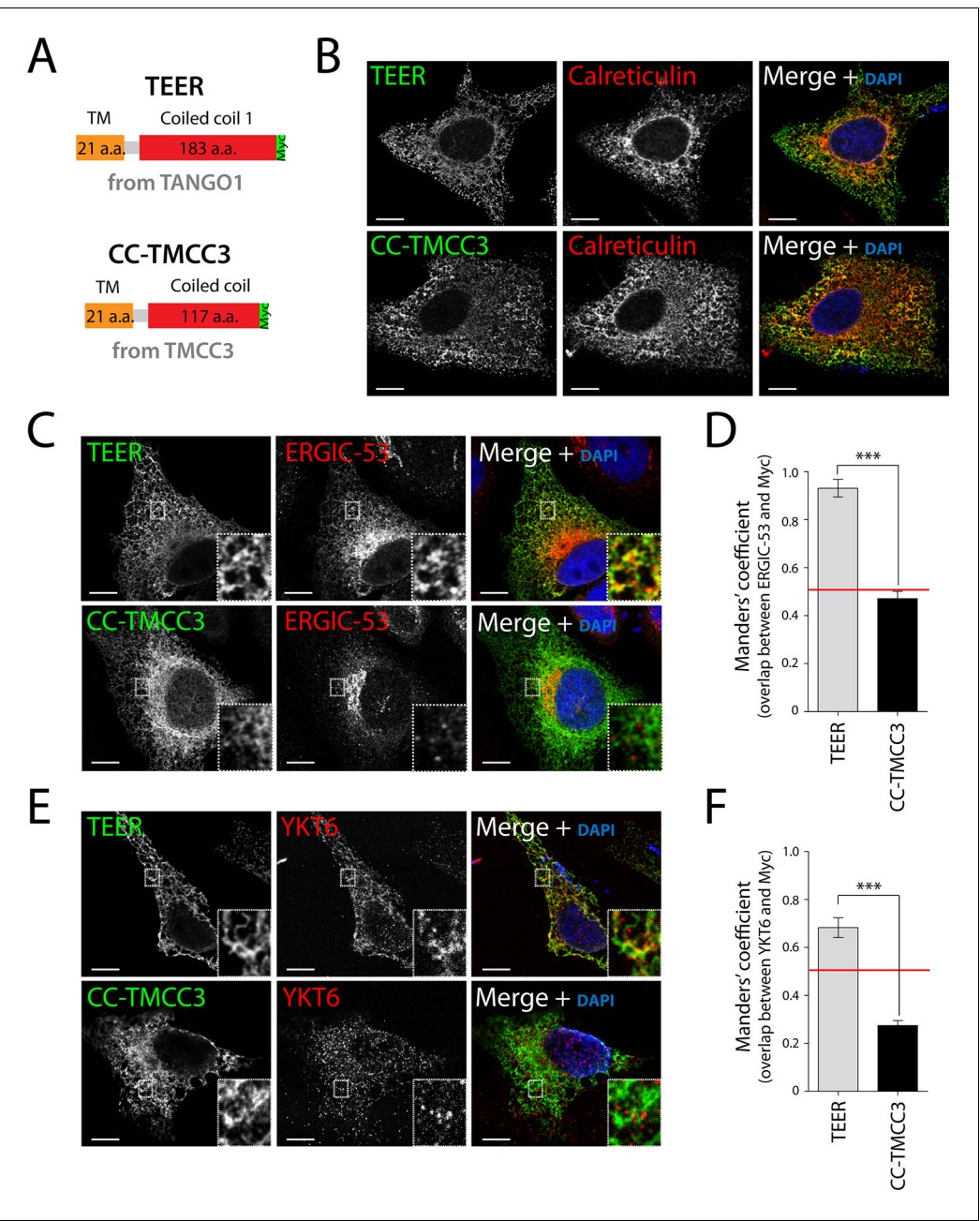

**Figure 3.** Expression of TEER recruits YKT6-containing ERGIC membranes to the ER. (**A**) A scheme depicting the coding regions of the two constructs - TEER-Myc (residues 1177–1396 of TANGO1, tagged with Myc) and CC-TMCC3-Myc (residues 282–437 of TMCC3, tagged with Myc). HeLa cells were seeded on coverslips and fixed 48 hr after transfection with the constructs. For visualization, upon fixation and permeabilisation, cells were stained with DAPI (blue) and immunostained for Myc (green). The red channel shows immunostaining for (**B**) the ER marker calreticulin; for (**C**) the transmembrane protein ERGIC-53; and (**E**) the v-SNARE YKT6 (cells were washed after permeabilisation to remove the cytosolic proteins prior to fixation as described in materials and methods). In the case of CC-TMCC3, YKT6 signal intensity was increased 3 times during image acquisition. Zooms are of 5x and scale bars correspond to 10 µm. (**D-F**) Colocalisation quantifications were calculated by Manders' correlation coefficient by measuring the overlap between the green and the red channels. Error bars reflect the standard error of the mean (SEM) of more than 20 cells from at least three independent experiments. ***p<0.001.

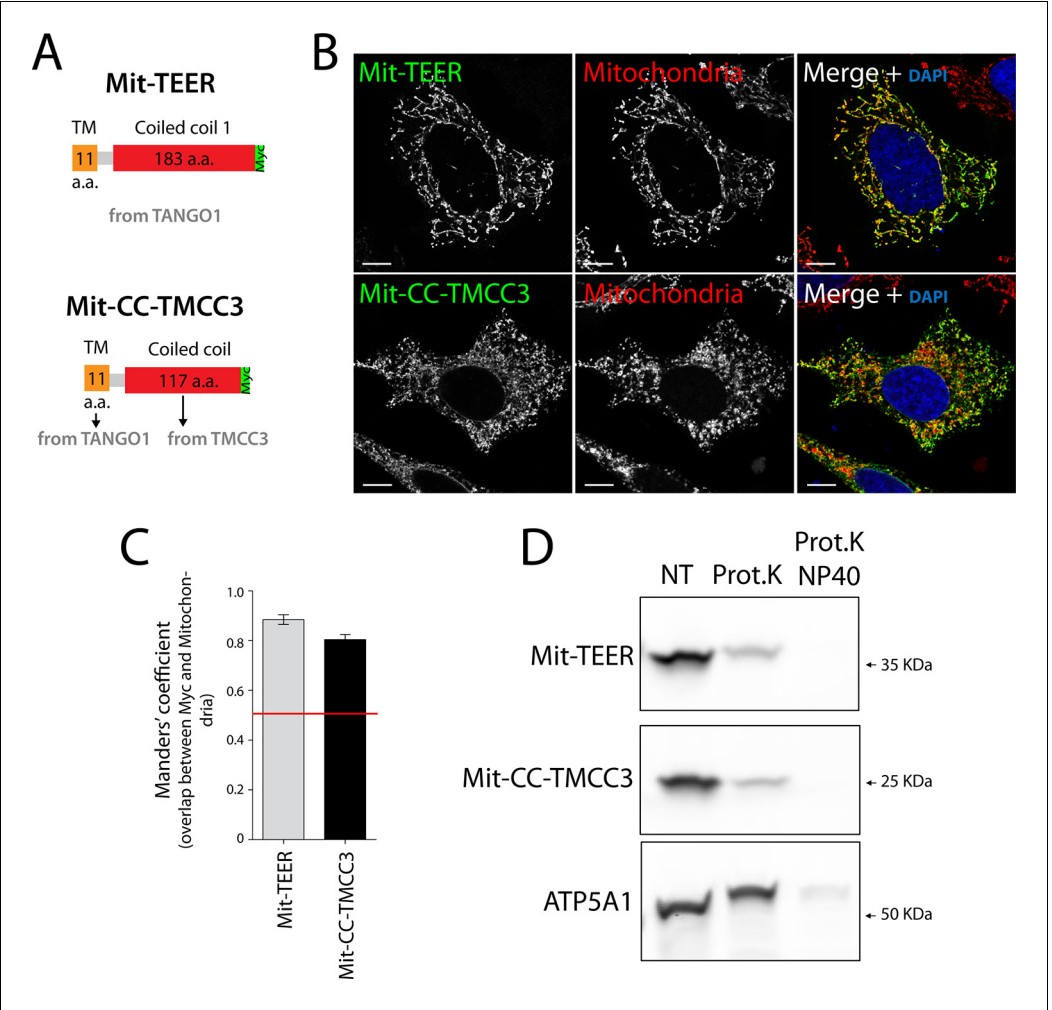

**Figure 4.** The sequence and topology of mitochondrially targeted TEER (Mit-TEER) and CC-TMCC3 (Mit-CC-TMCC3). (**A**) A scheme depicting the coding regions of the two constructs for mitochondrial targeting - Mit-TEER-Myc (residues 1187–1396 of TANGO1, including 11 amino acids of its transmembrane domain, tagged with Myc) or Mit-CC-TMCC3-Myc (the same 11 amino acids of TANGO1's transmembrane domains and residues 282–398 of TMCC3, tagged with Myc). (**B**) HeLa cells were seeded on coverslips and transfected with Mit-TEER-Myc or Mit-CC-TMCC3-Myc for 48 hr. To visualize mitochondria (red), cells were either incubated with the Mito-tracker dye for 30 min in serum-free medium before fixation or stained with an anti-ATP5A1 antibody after fixation and permeabilisation. They were also stained with DAPI (blue) and immunostained for Myc (green). (**C**) Colocalisation quantifications were calculated by Manders' correlation coefficient by measuring the overlap between the green and the red channels. Error bars reflect the standard error of the mean (SEM) of more than 20 cells from at least three independent experiments. Scale bars correspond to 10 μm. ***$P$ <0.001. (**D**) Mitochondrial membranes were isolated from HeLa cells transfected with either Mit-TEER or Mit-CC-TMCC3 and split into three fractions. The first was kept as a control (NT); proteinase K (0.1mg/ml) (Prot.K) was added to the second fraction; and to the third fraction, both proteinase K and the detergent NP40 (at a final concentration of 1%) were added (Prot.K/NP40). Samples were incubated on ice for one hour and western blotted with an anti-Myc antibody, to detect Mit-TEER and Mit-CC-TMCC3, and an anti-ATP5A1, a subunit of the ATP synthase, localized to the mitochondrial inner membrane. The loss of Mit-TEER/Mit-TMCC3 but not of ATP5A1, in the absence of detergent, indicated that these constructs are exposed to the cytoplasm.

## TANGO1 recruits ERGIC membranes for fusion with the ER

Blocking the exit of procollagen VII from the ER by temperature block and subsequent release or by knocking down SNAREs involved in procollagen VII export from the ER, revealed accumulation of ERGIC-53-containing membranes closely apposed to patches of procollagen-enriched ER

membranes. However, knockdown of TANGO1 dissociated ERGIC membranes from these patches (*Figure 2*), strongly indicating the involvement of TANGO1 in this recruitment. We mapped this function to amino acids 1214–1396 of TANGO1, corresponding to the first coiled coil domain in the cytoplasm. This "TEER" domain of TANGO1, when expressed at the surface of mitochondria, was able to specifically recruit ERGIC-53 membranes, but not COPII components or membranes representing early or late Golgi cisternae (*Figure 5* and *Figure 6*). We cannot rule out the possibility that other portions of TANGO1 coordinate with TEER domain to modulate the overall process of ERGIC recruitment.

ERGIC membranes recruited by TANGO1 contain the v-SNARE YKT6, depletion of which results in a block of procollagen VII export from the ER. In addition, ER-localised t-SNAREs (STX18, USE1 and BNIP1) are also required for this export, providing compelling evidence that membrane fusion with the ER is necessary for procollagen VII export (*Figure 1*). The fact that not all ERGIC-53 membranes are recruited by TANGO1's TEER domain and a specific set of SNAREs are required for fusion with the ER provides an explanation for our observation that TANGO1 knockdown does not alter general ER protein export or retrograde traffic from Golgi to the ER (*Nogueira et al., 2014*; *Saito et al., 2009*).

## A model for the export of procollagen VII from the ER

We propose the following pathway by which mega-carriers form at the ER for procollagen VII export (*Figure 6—figure supplement 2*):

First, TANGO1 binds procollagen VII in the ER lumen by a process that is dependent on its SH3-like domain. Second, the first coiled coil domain of cTAGE5 recruits the Sar1-specific, guanyl nucleotide exchange factor, Sec12 (*Saito et al., 2014*), which we suggest recruits more Sec23/Sec24 to increase the size of the inner COPII shell at ER exit sites. TANGO1's second coiled coil domain binds cTAGE5 via its second coiled coil domain, and together the PRD's of these proteins interact with Sec23/Sec24 (*Saito et al., 2009*; *Saito et al., 2011*). PRD binding to Sec23/Sec24 may expose the TEER domain to recruit or tether YKT6-containing ERGIC membranes that fuse with the procollagen VII-enriched ER domain.

In sum, the mechanism by which a procollagen VII-containing carrier grows depends on TANGO1-mediated capture and concentration of the cargo in the lumen which, coupled to its binding to Sec23/24, exposes the TEER domain to recruit a pool of ERGIC-53 membranes. The fusion of ERGIC membranes to procollagen VII-enriched domains provides a means for the growth of a mega export carrier commensurate with cargo size. This process of carrier biogenesis is thus fundamentally different from the mechanism by which generic COPII vesicles form at the ER to transport secretory cargoes.

There are approximately 40,000 TANGO1 molcules per cell (*Kulak et al., 2014*) and the number of ER exit sites is estimated to be a few hundred (*Hammond and Glick, 2000*). This suggests 100–200 TANGO1-cTAGE5 dimers at each exit site and if each TANGO1 (cTAGE5 does not contain a procollagen binding site in the lumen of the ER) binds a procollagen trimer, this would pack 100–200 fully folded trimers of procollagen into a mega-carrier at each ER exit site. We suggest that after packing approximately this quantity of procollagen, the separation of the collagen-filled carrier is triggered by dissociation of TANGO1 from procollagen VII and its concomitant detachment from Sec23/Sec24. Sec13/Sec31 would then be recruited to Sec23/Sec24 at the neck of the container filled with procollagen VII for its movement further along the secretory pathway (*Figure 6—figure supplement 2*).

## Materials and methods

### cDNA cloning and constructs

TEER constructs were amplified from TANGO1 constructs described previously (*Saito et al., 2009*). The CC-TMC33 constructs were amplified by RT-PCR from total mRNA of HeLa cells. All constructs were cloned into the vector pcDNA3.1 with a myc-his tag at the C-terminus (Invitrogen, Life Technologies). SnapGene software (from GSL Biotech, Chicago, IL; available at www.snapgene.com) was used for molecular cloning design. TANGO1/MIA3 CRISPR/Cas9 knockout plasmids (cat # sc-

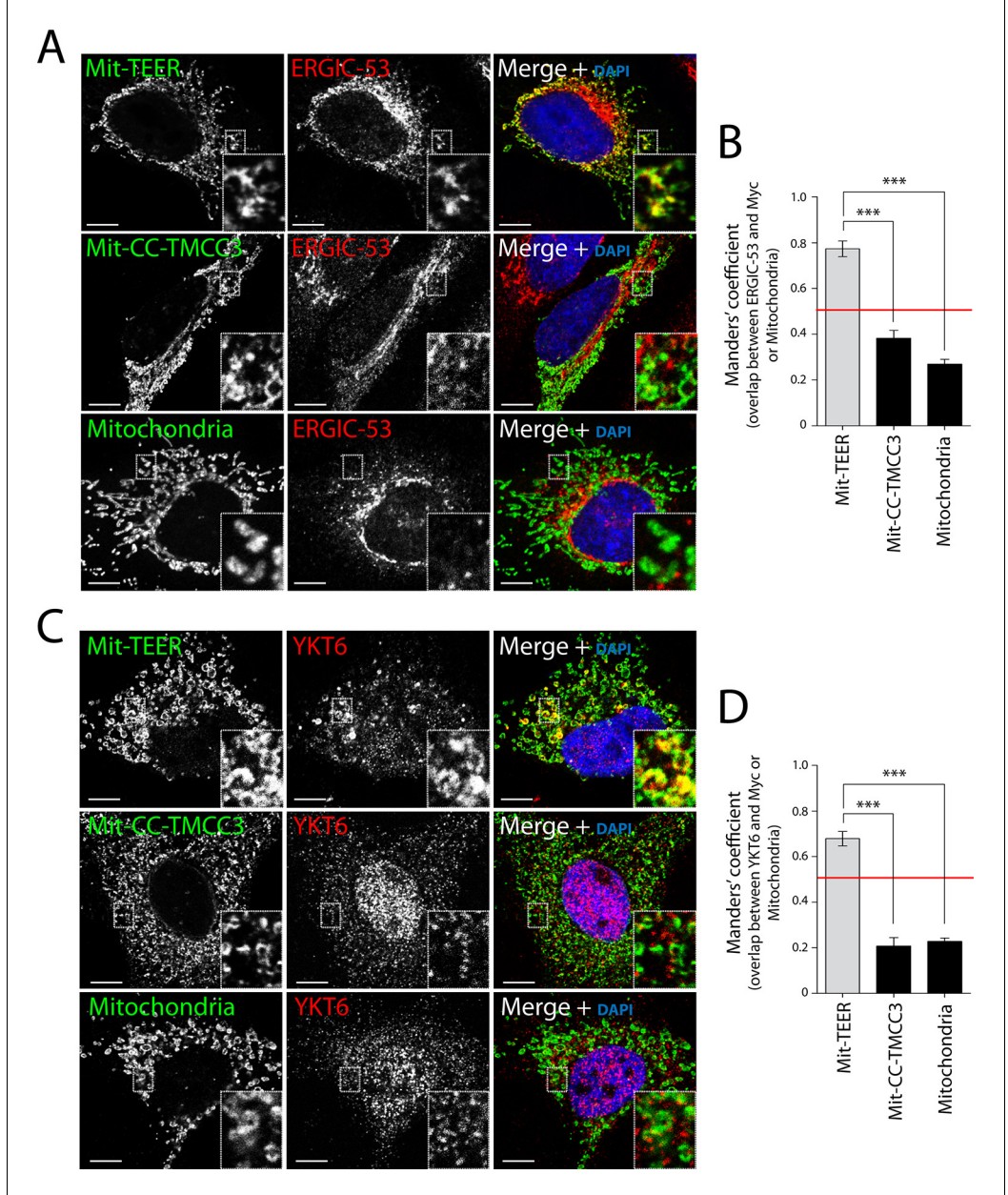

**Figure 5.** Recruitment of YKT6-containing ERGIC membranes to TEER expressed at the mitochondria (Mit-TEER). HeLa cells were seeded on coverslips and transfected with Mit-TEER-Myc or Mit-CC-TMCC3-Myc for 48 hr. For visualisation, cells were stained with DAPI (blue) and immunostained for Myc (green), and were also immunostained for (**A**) the transmembrane protein ERGIC-53 (red); and (**C**) the v-SNARE YKT6 (red) (in cells washed after permeabilisation to remove cytosolic protein followed by fixation and immunostaining). In the case of CC-TMCC3 and untransfected control, YKT6 signal intensity was increased 3 times during image acquisition. Zooms are of 5x and scale bars correspond to 10 µm. (**B-D**) Quantification of colocalisation was calculated by Manders' correlation coefficient by measuring the overlap between the green and the red channels. Error bars reflect the standard error of the mean (SEM) of more than 20 cells counted from at least three independent experiments. ***$P < 0.001$.

403994), consisting of a pool of three plasmids each encoding the Cas9 nuclease and a TANGO1/MIA3-specific 20 nt guide RNA (gRNA), were obtained from Santa Cruz Biotechnology (Dallas, USA).

## Cell culture
RDEB/FB/C7 cells, HeLa, and Het-1A cells were grown at 37°C with 5% $CO_2$ in complete Dulbecco's Modified Eagle Medium (DMEM) with 10% fœtal bovine serum unless otherwise stated.

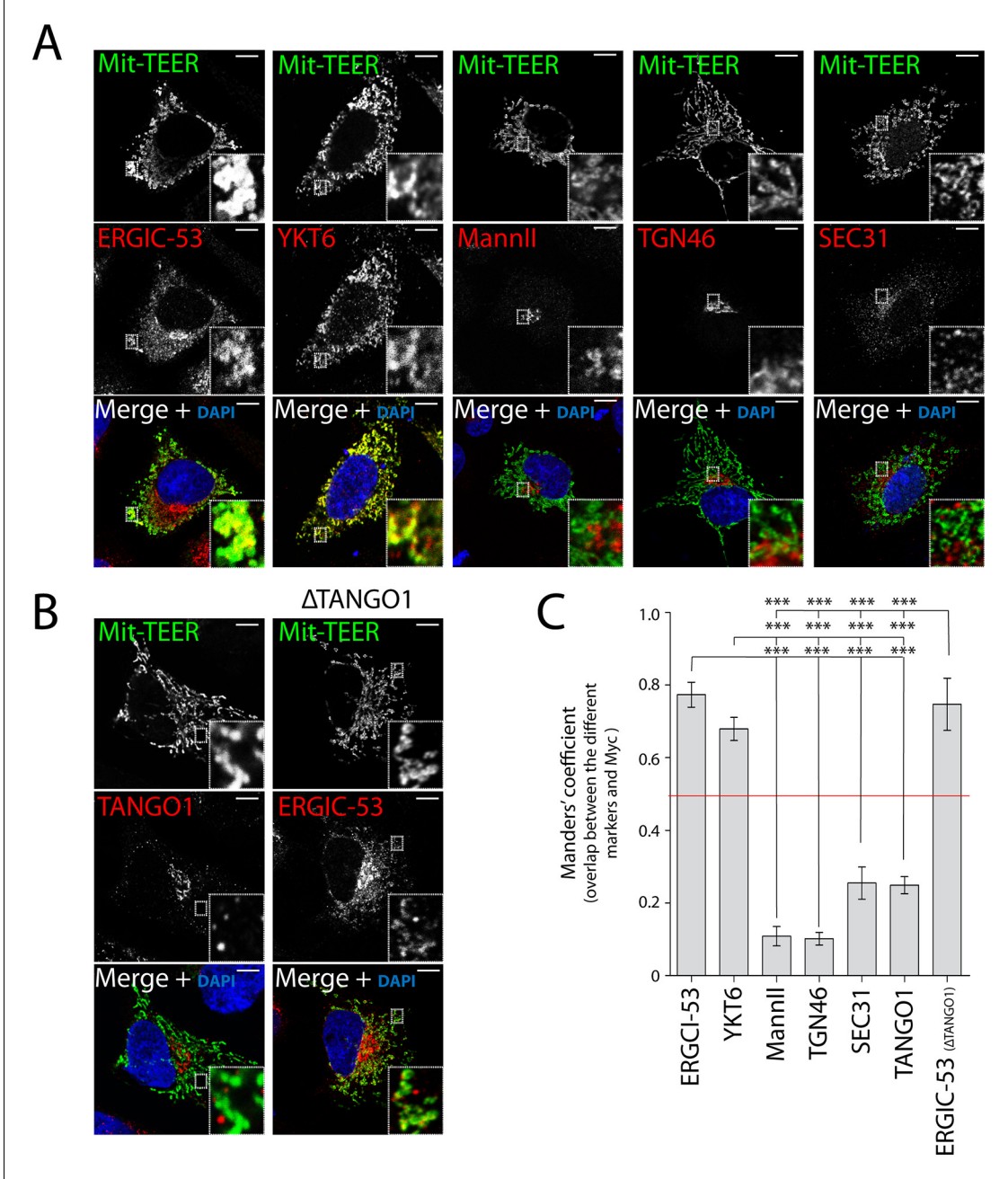

**Figure 6.** TEER recruits ERGIC membranes, but not Golgi membranes or COPII components HeLa cells were seeded on coverslips and 48 hr after transfection with Mit-TEER-Myc they were fixed and permeabilised. (**A**) For visualisation, cells were stained with DAPI (blue) and immunostained for Myc (green). The red channel shows immunostaining for ERGIC-53; YKT6 (after cytosol washout); early and late Golgi markers (Mannosidase II and TGN46); and the COPII coat component Sec31. (**B**) Left panel - HeLa cells transfected with Mit-TEER-Myc as before were visualised following DAPI staining (blue), immunostaining for Myc (green) and TANGO1 (red). Right panel - HeLa cells depleted of TANGO1 protein by CRISPR procedure were transfected with Mit-TEER-Myc as before and visualized following DAPI staining (blue) and immunostaining for Myc (green) and ERGIC-53 (red). Zooms are of 5x and scale bars correspond to 10 μm. (**C**) Quantifications of colocalisation was calculated by Manders' correlation coefficient by measuring the overlap between the green and the red channels. Error bars reflect the standard error of the mean (SEM) of more than 10 cells counted from at least three independent experiments. ***$P$ <0.001.

The following figure supplements are available for figure 6:

**Figure supplement 1.** Generation of TANGO1-deficient HeLa by CRISPR/Cas9.

*Figure 6 continued on next page*

*Figure 6 continued*

**Figure supplement 2.** A model for the generation of procollagen VII containing mega-carriers.

Plasmids were transfected in HeLa, with TransIT-HeLa Monster (Mirus Bio LLC, Madison, WI) according to the manufacturer's protocols. siRNAs were transfected in RDEB/FB/C7 and Het-1A with Hiperfect (Qiagen, Venlo, Netherlands) according to the manufacturer's protocols. In the case of RDEB, siRNAs were transfected twice (both times in suspension), on day 0 and on day 1. siRNA transfections in Het-1A were also performed twice, once in suspension on day 0 and then on adherent cells on day 1.

## siRNA oligos
siRNA oligos for TANGO1, STX18, BNIP1, USE1, YKT6, SEC22B, BET1 were purchased from Eurofins MWG Operon (Huntsville, AL): TANGO1 siRNA- GAUAAGGUCUUCCGUGCUU; STX18 CAGGACC-GCUGUUUUGGAUUU; YKT6 GGAGAAGGUACUAGAUGAA or GAAGGUACUAGAUGAAUUC; BNIP1 CCAAAGAGAGCCUGGCCCA; BET1 GCUGCUGUGCUAUAUGAUG; USE1 GGUGAUCAAU-GAAUAUUCC; Control siRNAs consisted of a pool of ON-TARGETplus Non-Targeting siRNAs (D-001810-10-05, Thermo Scientific, Waltham, MA).

## Measuring gene expression by quantitative RT-PCR
RDEB/FB/C7 and Het-1A cells were lysed and total RNA was extracted with the RNeasy extraction kit (Qiagen, Netherlands). cDNA was synthesized with Superscript III (Invitrogen). To determine expression levels of BNIP1, USE1 and BET1, quantitative real-time PCR was performed with Light Cycler 480 SYBR Green I Master (Roche, Switzerland) according to manufacturer's instructions.

## Antibodies
Antibodies used in western blotting and immunocytochemistry; Col VII and ATP5A1 (Abcam, Cambridge UK), Syntaxin 18 (Santa Cruz Biotechnology, Dallas, USA), ERGIC-53 (Santa Cruz Biotechnology, Dallas, USA; Enzo Life Sciences, Farmingdale, New York, USA), Sec31A (BD Biosciences), TANGO1, YKT6, c-Myc (9E10), c-Myc and alpha-tubulin (SIGMA-Aldrich, St. Louis, MO, USA), Mannosidase II (Merck, Kenilworth, NJ, USA), TGN46 (abd serotec, Kidlington, UK), HSP47 and Calreticulin (Enzo Life Sciences, Farmingdale, New York).

## Immunofluorescence microscopy
Cells grown on coverslips were fixed either with cold methanol for 10 min at −20°C or with 4% formaldehyde in PBS for 10 min, followed by permeabilisation with 0.2% Triton X100 at room temperature, and then incubated with blocking reagent (Roche, Basel, Switzerland) for 30 min at room temperature. Primary antibodies were diluted in blocking reagent and incubated overnight at 4°C. Secondary antibodies conjugated with Alexa 488, Alexa 594 or Alexa 647 were diluted in blocking reagent and incubated for 1 hr at room temperature. For YKT6 localisation studies, HeLa cells were transfected with the indicated constructs; after 48 hr after transfection, cells were processed for immunofluorescence. Briefly, cells were washed twice with room temperature KHM buffer (125 mM potassium acetate, 25 mM HEPES [pH 7.2], and 2.5 mM magnesium acetate). Cells were permeabilised by incubation in KHM with 0.1% Saponin for 5 min on ice followed by washing for 7 min on ice with KHM buffer. Cells were subsequently fixed in 4% paraformaldehyde and processed for immunofluorescence microscopy. For staining of mitochondria, before fixation, cells were incubated with a final concentration of 200 nM of MitoTracker (Invitrogen, Carlsbad, CA, USA) for 30 min in serum-free medium or, after fixation and permeabilisation immunostaining was carried out with anti-ATP5A1. Images were taken with a Leica TCS SPE or Leica TCS SP5 confocal with a 63x objective. Two-channel colocalisation analysis was performed using ImageJ, and the Manders' correlation coefficient was calculated using the plugin JaCop (*Bolte and Cordelieres, 2006*).

## Collagen secretion

The secretion assay was carried out as described previously (*Nogueira et al., 2014*). Briefly RDEB/FB/C7 cells were transfected with siRNA as described above. 24 hr after the last siRNA transfection, the medium was replaced with a fresh medium containing 2 µg/ml ascorbic acid. Medium with ascorbate was added to the cells for 20 hr to allow for collagen secretion. The media were centrifuged at low speed to remove any cells or cellular debris and then the supernatant was boiled for 5 min with Laemmli SDS-sample buffer. For cell lysis, the cells were washed with PBS, lysed and centrifuged at 14,000 rpm for 15 min at 4°C. The supernatants were boiled for 5 min with Laemmli SDS-sample buffer. Both media and cell lysate were subjected to SDS-PAGE (6% acrylamide) and western blotting with Collagen VII, SEC31 and Tubulin antibodies. ImageJ (NIH, Bethesda, Maryland) was used for quantification. Values plotted are a ratio of the signal obtained from the intensities of bands of collagen VII in the medium to those in the corresponding lysate to obtain a quantitative measure of collagen secretion free from variations that arise due to differences in cell densities. A similar procedure was used to measure secretion from HeLa cells that were transfected with Collagen VII-FLAG. Briefly, control or CrispR HeLa cells lacking TANGO1 (∆TANGO1) were transfected with a FLAG-epitope-tagged collagen VII plasmid. 28 hr later, cells were washed and changed into medium with ascorbate for 20 hr. The cell lysate and medium were processed to assess secretion of Collagen VII as described for RDEB/FB/C7 cells.

## Protease protection assay to determine the topology of Mit-TEER and Mit-CC-TMCC3

Crude mitochondrial-membrane isolation was carried out from HeLa cells transiently transfected with either Mit-TEER or Mit-CC-TMCC3 by mechanical lysis of transfected cells from one to two 10 cm culture dishes, spinning at 700 g to remove intact cells and nuclei and then spinning at 7000 g for 10 min to obtain a pellet substantially enriched in mitochondrial membranes. These membranes were split into three fractions. The first was kept as a control; to the second fraction was added proteinase K (0.1 mg/ml); and to the third fraction, we added both proteinase K as well as the detergent NP40 (at a final concentration of 1%). Samples were incubated on ice for one hour to allow proteinase K to digest all accessible protein. The reaction was terminated by the addition of PMSF (1 mM) for ten minutes and then immediate boiling at 100°C for ten minutes along with acrylamide gel loading buffer. Western blotting was performed on these samples using an anti-Myc antibody and an antibody against the mitochondrial matrix protein ATP5A1 (a subunit of the ATP synthase, localised to the mitochondrial inner membrane). We compared the accessibility of the respective proteins to proteinase K in the presence and absence of detergent. Loss of a Mit-TEER/Mit-CC-TMCC3 in the absence of detergent would indicate that the protein was exposed to the cytoplasm. The mitochondrial inner membrane protein ATP5A1 served as a control to confirm that mitochondrial membranes were intact at the outset of the experiment and remained so in the absence of detergent (NP40).

## Generation of TANGO1 knockout HeLa cell line by CRISPR

To generate a HeLa cell line in which TANGO1 protein expression has been abolished, we performed genome editing with the clustered regularly interspaced short palindromic repeat (CRISPR)-Cas9 system. We obtained a pool of three plasmids each containing a 20 nt guide RNA (gRNA) sequence designed to target double-strand breaks in the Tango1 coding sequence and the pSpCas9 ribonuclease (Santa Cruz, sc-403994). In addition, plasmids contained the GFP coding sequence to allow for positive selection of transfected cells. HeLa cells were transfected with 1µg of pooled plasmid using the TransIT HeLa MONSTER reagent (Mirus Bio LLC). 96 hr after transfection GFP-positive cells were isolated by fluorescence-activated cell sorting, and single cells were collected in 96-well plates. After expansion till 6-well format, cells were collected and protein lysates were prepared to assess TANGO1 presence by immunoblotting.

## Statistical analysis

Results shown are mean ± standard error of the mean (SEM). Statistical testing was performed using Student's *t*-test (continuous data, two groups). The Student's *t*-test was performed after a one-way ANOVA for the comparison of more than 2 groups. *p<0.05; **p<0.01; ***p<0.001. For immunofluorescence microscopy analysis, the number of cells was greater than 10. The number of experiments

was greater than three for each quantification. Statistical analyses to compare the differences between densitometric measurements in collagen VII secretion was carried out using a two-tailed Mann Whitney U test.

## Acknowledgements

We thank members of the Malhotra lab, particularly Cristina Nogueira and Patrik Erlmann for valuable discussions. All confocal imaging was performed in the Centre for Genomic Regulation Advanced Light Microscopy Unit.

We acknowledge support of the Spanish Ministry of Economy and Competitiveness, 'Centro de Excelencia Severo Ochoa 2013-2017', SEV-2012-0208. The research leading to these results has received funding from the European Union Seventh Framework Programme (FP7/2007-2013) under grant agreement n° 625149 to AJM Santos and ERC grant agreement n°609989 to V Malhotra. This work reflects only the author's views and the Community is not liable for any use that may be made of the information contained therein.

V Malhotra is an Institució Catalana de Recerca i Estudis Avançats professor at the Centre for Genomic Regulation, and the work in his laboratory is funded by grants from MINECO's Plan Nacional (Ref. BFU2013-44188-P) and Consolider (CSD2009-00016).

## Additional information

### Competing interests

VM: Senior editor, *eLife.* The other authors declare that no competing interests exist.

### Funding

| Funder | Grant reference number | Author |
| --- | --- | --- |
| European Commission | European Union Seventh Framework Programme (FP7/2007-2013) 625149 | António JM Santos |
| European Research Council | 609989 | Ishier Raote Margherita Scarpa Nathalie Brouwers Vivek Malhotra |
| Ministerio de Economía y Competitividad | SEV-2012-0208 | Vivek Malhotra |
| Ministerio de Economía y Competitividad | Plan Nacional BFU2013-44188-P | Vivek Malhotra |
| European Research Council | Consolider CSD2009-00016 | Vivek Malhotra |

The funders had no role in study design, data collection and interpretation, or the decision to submit the work for publication.

### Author contributions

AJMS, IR, Conception and design, Acquisition of data, Analysis and interpretation of data, Drafting or revising the article, Contributed unpublished essential data or reagents; MS, Acquisition of data, Contributed unpublished essential data or reagents; NB, Acquisition of data, Analysis and interpretation of data, Contributed unpublished essential data or reagents; VM, Conception and design, Analysis and interpretation of data, Drafting or revising the article

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
