## [Decision Letter]

Thank you for submitting your work entitled "TANGO1 recruits ERGIC membranes to the Endoplasmic Reticulum for procollagen export" for peer review at *eLife*. Your submission has been favorably evaluated by Randy Schekman (Senior editor) and three reviewers, one of whom is a member of our Board of Reviewing Editors. The reviewers have discussed the reviews with one another and the Reviewing editor has drafted this decision to help you prepare a revised submission.

The goal of a Research Qdvance is to publish experiments that build on work in an important way that has been published by *eLife* as a Research Article. This manuscript fits this description and will be of broad interest to the readers of *eLif*e after the authors address the following concerns.

Previous papers have shown that TANGO1 drives collagen secretion from the ER and works together with cTAGE and COPII coat proteins in this process. In a previous *eLife* paper, these authors found that the ER t-SNARE, STX18 is also needed. Here they reveal the full complement of SNARE proteins needed and report that a domain of TANGO1 may help tether ERGIC membranes to the ER as part of the present process. If the authors can address all the following comments, the work should be published in *eLife* as an Advance.

1) The weakest part of the otherwise strong story relates to the conclusion that CC1 is sufficient to tether ERGIC membranes and drive retargeting of TANGO CC1 to mitochondria. First, the authors use a transmembrane domain at the N-terminus of CC1 that is normally present in the middle of the protein to test possible tethering function of CC1. Please verify in some manner that this really acts as a tail anchor, leaving the CC1 facing the cytosol. Next, the authors cut the same domain in half and it goes to mitochondria, even though one reviewer thought that mitochondrial targeting signals are amphipathic helices. Please add quantitation of the overlap between an ER marker and the mitoTEER construct. Also, how much ERGIC overlap is seen with mitotracker in cells not expressing mitoTEER? This is important because it impacts the significance of mitoTEER retargeting of ERGIC in their experiments. There are more and more reports of close interaction between ER and mitochondria that are important for lipid exchange. Please look closely at this section and address this with careful quantitation. Also, the mitochondrial experiment obviously needs a control construct to be analyzed under identical conditions. A Mit-CC3 construct seems a good choice.

2) The results of the assay in Figure 3 need to be quantified and it should be done by an objective measure using image analysis tools. This will mean thinking about what constitutes recruitment in the assay. How close does an ERGIC spot need to be? How many per collagen structure? What fraction of total ERGIC spots is recruited?

3) Please quantify the overlap between ERGIC53 and YKT6. Also, please indicate mobility of size markers on all blots.

4) In the last paragraph of the subsection “TANGO1 recruits ERGIC membranes for fusion with the ER”, how clear is it that it is a subpopulation of ERGIC53 membranes that are recruited? Please clarify with quantitation.

5) The effects shown in Figure 4 are very modest, and it would be informative to know what the Pearson's value is for a mock-transfected (vector only) control. Furthermore is a difference between 0.5 and 0.65 Pearson's biologically relevant? I think the authors should change their description of the data in Figure 4 (subsection “A domain in TANGO1 directly recruits ERGIC membranes”). However, the data shown in Figure 4 are much more convincing, indicating that a sub-population of ERGIC compartments (YKT6 positive) are indeed recruited, as the authors eventually discuss. Therefore, I agree with the overall interpretation but I was put off by the overstatement regarding Figure 4.

6) Is collagen secretion blocked in the tango1-CRISPR cells? This seems like an important control for Figure 5.

7) Figure 3 uses quadruple labeling (all in one panel) to compare ERGIC/collagen juxtaposition between control and TANGO1 knockdown cells. It is really difficult to assess the result. Side-by-side presentation of single channel images is needed (as in Figure 2).

8) The idea seems to be that Mit-TEER repositions the entire ERGIC53 (or YTKT6) pattern so that it aligns with the mitochondrial pattern. A larger field view (in addition to the enlarged insets) would probably convey this point well, particularly if evident for TEER but not CC3.

9) In the previous study, the authors showed that knockdown of SLY1 and Syntaxin 18 did not significantly affect the ER-Golgi recycling pathway or secretion of a different collagen. The authors appear to be assuming that knockdown of USE1, BNIP1, and YKT6 are similarly specific, but they did not actually demonstrate this. Are they? Please clarify in the text.

10) There is a typo in the third paragraph of the subsection “A model for the export of procollagen VII from the ER”: "Sec12-GTP" should be "Sar1-GTP". It was not clear to me why, as described in the next paragraph, binding of Tango1 to Sec23/24 would "expose the TEER domain". Is there any evidence that the TEER domain is normally masked?

[Editors' note: further revisions were requested prior to acceptance, as described below.]

Thank you for resubmitting your work entitled "TANGO1 recruits ERGIC membranes to the endoplasmic reticulum for procollagen export" for further consideration at *eLife*. Your revised article has been favorably evaluated by Randy Schekman (Senior editor) and a Reviewing editor. The paper has been greatly improved by significant rewriting and inclusion of new data and better quantification. We favor publication of the story after correction of three minor items.

1) Ykt6 staining is weak in Figure 3 and Figure 5.

2) Although you have shown that the TEER domain is sufficient to recruit ERGIC to mitochondria, you have not yet shown that other regions of TANGO don't also have this capacity. Please state this explicitly.

3) The ERGIC vesicles in Figure 6—figure supplement 2 are likely larger than those shown. If it is not too difficult, it would be worth trying to make the various compartments of appropriate scale.

---

## [Author Response]

*[…] 1) The weakest part of the otherwise strong story relates to the conclusion that CC1 is sufficient to tether ERGIC membranes and drive retargeting of TANGO CC1 to mitochondria. First, the authors use a transmembrane domain at the N-terminus of CC1 that is normally present in the middle of the protein to test possible tethering function of CC1. Please verify in some manner that this really acts as a tail anchor, leaving the CC1 facing the cytosol.*

To address this, we have performed a protease protection assay using a crude mitochondrial membrane preparation from cells transfected with either the CC1 (TEER) construct or another control coiled coil construct (Mit-CC-TMCC3). We compared the accessibility of the respective proteins to proteinase K in the presence and absence of detergent. A near-complete loss of a Mit-TEER/Mit-CC-TMCC3 in the absence of detergent confirmed that the protein was exposed to the cytoplasm. The mitochondrial inner membrane protein ATP5A1 served as a control to confirm that mitochondrial membranes were intact at the outset of the experiment and remained so in the absence of detergent (NP40). This data is now included in the manuscript, Figure 4.

Next, the authors cut the same domain in half and it goes to mitochondria, even though one reviewer thought that mitochondrial targeting signals are amphipathic helices. Please add quantitation of the overlap between an ER marker and the mitoTEER construct. Also, how much ERGIC overlap is seen with mitotracker in cells not expressing mitoTEER? This is important because it impacts the significance of mitoTEER retargeting of ERGIC in their experiments. There are more and more reports of close interaction between ER and mitochondria that are important for lipid exchange. Please look closely at this section and address this with careful quantitation. Also, the mitochondrial experiment obviously needs a control construct to be analyzed under identical conditions. A Mit-CC3 construct seems a good choice.

We have added a quantification of the overlap between an ER marker calreticulin and Mit-TEER in the text (Results, last paragraph).

As suggested, we have generated a similar, mitochondrially targeted control coiled coil construct, Mit-CC-TMCC3 (shown in Figure 4). We have confirmed its localization to the mitochondria and, as described above, shown that it is exposed to the cytoplasm (Figure 4). These data as well as a comparative quantification of the extent of overlap of ERGIC and mitochondria in cells transfected with Mit-TEER or cells with Mit-CC-TMCC3 is shown in Figure 5.

2) The results of the assay in Figure 3 need to be quantified and it should be done by an objective measure using image analysis tools. This will mean thinking about what constitutes recruitment in the assay. How close does an ERGIC spot need to be? How many per collagen structure? What fraction of total ERGIC spots is recruited?

We have added a quantitative measure of the recruitment of ERGIC to collagen patches (Figure 2).

3) Please quantify the overlap between ERGIC53 and YKT6. Also, please indicate mobility of size markers on all blots.

Done.

4) In the last paragraph of the subsection “TANGO1 recruits ERGIC membranes for fusion with the ER”, how clear is it that it is a subpopulation of ERGIC53 membranes that are recruited? Please clarify with quantitation.

As is clearly visible in the images we show describing ERGIC recruitment to the mitochondria, there is a substantial pool of ERGIC membranes that remain unaffected and are not mis-targeted. This is the basis for our statement that a subpopulation of ERGIC is recruited. Importantly, every cell that expresses Mit-TEER shows a distinct recruitment of ERGIC membranes, quantification of the fraction of ERGIC recruited will be misleading as it will depend on a number of factors such as expression levels of the Mit-TEER. To avoid confusion, we have changed the text and do not refer to it as a subpopulation of ERGIC membranes.

5) The effects shown in Figure 4 are very modest, and it would be informative to know what the Pearson's value is for a mock-transfected (vector only) control. Furthermore is a difference between 0.5 and 0.65 Pearson's biologically relevant? I think the authors should change their description of the data in Figure 4 (subsection “A domain in TANGO1 directly recruits ERGIC membranes”). However, the data shown in Figure 4 are much more convincing, indicating that a sub-population of ERGIC compartments (YKT6 positive) are indeed recruited, as the authors eventually discuss. Therefore, I agree with the overall interpretation but I was put off by the overstatement regarding Figure 4.

Instead of using only a TEER constructs (Mit-TEER), we now added a coiled coil-containing construct TMCC3 (Mit-CC-TMCC3) as a control for our observations. We now make comparisons between mitochondrial TEER and TMCC3 expressing cells. Adding more experiments to our analysis has resulted in an even more reliable and enhanced difference in the overlap coefficients. This is now included in the text.

6) Is collagen secretion blocked in the tango1-CRISPR cells? This seems like an important control for Figure 5.

Yes. We have included this control (Figure 6—figure supplement 2).

7) Figure 3 uses quadruple labeling (all in one panel) to compare ERGIC/collagen juxtaposition between control and TANGO1 knockdown cells. It is really difficult to assess the result. Side-by-side presentation of single channel images is needed (as in Figure 2).

Done.

8) The idea seems to be that Mit-TEER repositions the entire ERGIC53 (or YTKT6) pattern so that it aligns with the mitochondrial pattern. A larger field view (in addition to the enlarged insets) would probably convey this point well, particularly if evident for TEER but not CC3.

As mentioned above, there is a distinct ERGIC recruitment to Mit-TEER, but importantly, a large pool of ERGIC is clearly not mis-localised/repositioned. This is not seen with CC-TMCC3 domain. We have changed the images as required.

9) In the previous study, the authors showed that knockdown of SLY1 and Syntaxin 18 did not significantly affect the ER-Golgi recycling pathway or secretion of a different collagen. The authors appear to be assuming that knockdown of USE1, BNIP1, and YKT6 are similarly specific, but they did not actually demonstrate this. Are they? Please clarify in the text.

Based on the data obtained from TANGO1 it is very likely that this combination of SNAREs is likely involved in the export of procollagen VII. Other combinations might be used to transfer cargo between ER and the Golgi.

10) There is a typo in the third paragraph of the subsection “A model for the export of procollagen VII from the ER”: "Sec12-GTP" should be "Sar1-GTP". It was not clear to me why, as described in the next paragraph, binding of Tango1 to Sec23/24 would "expose the TEER domain". Is there any evidence that the TEER domain is normally masked?

We have rewritten the Discussion to better explain our model. Briefly, the process we describe must be regulated to prevent rampant retrograde fusion of ERGIC membranes with the ER. One reasonable way to regulate this would be to expose the CC1 domain of TANGO only when it is needed. This we propose might be triggered by the binding of PRD’s of TANGO-cTAGE5 to Sec23/Sec24 (Figure 9). Structural analysis of these components in the future would be real test of this working model.

[Editors' note: further revisions were requested prior to acceptance, as described below.]

1) Ykt6 staining is weak in Figure 3 and Figure 5.

We have included new images where, according to the editor’s suggestions, the gain for the YKT6 signal was boosted while acquiring images for control conditions and this has described in corresponding figure legends. This increase in gain has lead to a clearly visible staining in the nuclear area, probably due to the quality of the antibody. Regardless, we do not see any significant localization of YKT6 to mitochondria when cells express CC-TMCC3 or in untransfected cells.

2) Although you have shown that the TEER domain is sufficient to recruit ERGIC to mitochondria, you have not yet shown that other regions of TANGO don't also have this capacity. Please state this explicitly.

Done.

3) The ERGIC vesicles in Figure 6—figure supplement 2 are likely larger than those shown. If it is not too difficult, it would be worth trying to make the various compartments of appropriate scale.

Done.